# System Size Dependence in the Zimm–Bragg Model: Partition Function Limits, Transition Temperature and Interval

**DOI:** 10.3390/polym13121985

**Published:** 2021-06-17

**Authors:** Artem Badasyan

**Affiliations:** Materials Research Laboratory, University of Nova Gorica, Vipavska 13, SI-5000 Nova Gorica, Slovenia; artem.badasyan@ung.si or abadasyan@gmail.com

**Keywords:** Zimm–Bragg model, helix-coil transition, zipper model

## Abstract

Within the recently developed Hamiltonian formulation of the Zimm and Bragg model we re-evaluate several size dependent approximations of model partition function. Our size analysis is based on the comparison of chain length *N* with the maximal correlation (persistence) length ξ of helical conformation. For the first time we re-derive the partition function of zipper model by taking the limits of the Zimm–Bragg eigenvalues. The critical consideration of applicability boundaries for the single-sequence (zipper) and the long chain approximations has shown a gap in description for the range of experimentally relevant chain lengths of 5–10 persistence lengths ξ. Correction to the helicity degree expression is reported. For the exact partition function we have additionally found, that: at N/ξ≈10 the transition temperature $T_{m}$ reaches its asymptotic behavior of infinite *N*; the transition interval ΔT needs about a thousand persistence lengths to saturate at its asymptotic, infinite length value. Obtained results not only contribute to the development of the Zimm–Bragg model, but are also relevant for a wide range of Biotechnologies, including the Biosensing applications.

## 1. Introduction

The model suggested by Zimm and Bragg [1,2] in 1960s contains two size-related relevant scales: the degree of polymerization (DP) *N* which is related to the polypeptide chain length and the temperature-dependent correlation length ξ describing the scale of correlations in the secondary structure. In the absence of long-range interactions or under ideal conditions (in theta-point), the correlation length can be considered identical to the persistence length.

Speaking about size in terms of the Zimm–Bragg model, it is relevant to mention that the distance is measured in numbers of repeating units and the conformation is described with spins. Therefore, the questions about polymer coil size cannot be answered within the Zimm–Bragg model. In this respect it is similar to the Ising and Potts spin models [3], which also contain no measure of distance and do not describe the extension of a crystal.

While the Zimm–Bragg model is mostly used in its limiting, infinite size limit, recent advances in BioNanotechnologies trigger the need for a detailed consideration of chain length related effects. The manipulation of oligomers and short (2 to 10 persistence lengths) polymers requires distinct predictions about the secondary structure stability and cooperativity to enable the trustworthy comparison between the theoretical and experimental results.

We have recently shown [4] a successful application of the Zimm–Bragg model (in its single-sequence limit) to describe the single-strand DNA (ssDNA) adsorption on a carbon nanotube. ssDNA-carbon nanotube hybridization is a key to numerous biological [5] and biosensing [6] applications, and the quality of DNA adsorption on carbon nanotube essentially depends on the length of ssDNA [7]. Importantly, it was concluded that the adsorption of 6 repeat units long ssDNA oligomer on carbon nanotube is much more efficient, as compared to the ssDNA of 30 repeat units, without providing any physical explanation. While in the current study there is no intention to apply the Zimm–Bragg model to describe the adsorption of long ssDNA to CNT, better understanding of different limits of the model is a first step towards possible future developments in this direction.

Another recent study [8] has reported on Zimm–Bragg model application to describe the helix-coil transition in the classical experimental system of synthetic polypeptide homopolymers of different lengths. The compounds studied (PBLG) are ideal to test the validity of the theory, since the molecules are not globular, and the only relevant structures are helices. Nevertheless, the authors report serious disagreement: formulas with fixed values of degree of polymerization do not fit the experimental data. Thus the need for critical reconsideration of size-dependent properties of the Zimm–Bragg model becomes apparent.

Being one-dimensional model with short-range interactions, the model of Zimm and Bragg is one of few models in Physics, that allow for the exact analytical solution with well-defined partition function through the eigenvalues. The helicity degree naturally follows from the partition function derivative. Besides seldom used exact partition function, there are several size-dependent limits known, resulting in absolutely different expressions in terms of Zimm–Bragg model parameters. Thus when processing the experimental data and attempting the fit, the limits of applicability of different partition function approximations are crucial for the correctness of obtained results.

Recently suggested Hamiltonian formulation of the Zimm–Bragg model [9] offers straightforward computation of helicity degree from the exact partition function at any degree of polymerization *N*. Besides, it allows to derive different approximations of partition function in a straightforward and controllable manner and thus makes it easier to justify the limits of their applicability.

We re-derive the well-known approximations of the Zimm–Bragg model, thoroughly discuss the limits of their applicability and report the dependence of transition temperature and interval on degree of polymerization *N*.

## 2. Materials and Methods

### 2.1. Hamiltonian Formulation of the Zimm–Bragg Model

The microscopic formulation of the Zimm–Bragg model [9] makes use of a Potts-like formulation akin to the more general, one-dimensional many-body model [10], simplified to the nearest neighbor level. The same model is applicable to describe the conformations of double-stranded systems in the absence of large loops [11] and for the account of structural heterogeneity (heteropolymer) [12]. Since the most of applications of the Zimm–Bragg model are devoted to the polypeptide conformations and make use of a homopolymeric version of the model, I will undertake the same strategy.

Here we present the model and approach in brief and concentrate on the study of system size dependence of the Zimm–Bragg model and its different limits.

Assume Q(≥2) possible values for the spin $\gamma_{i}$ describing the state of the *i*-th repeated unit; $\gamma_{i}=1$ value corresponds to the hydrogen bonded, ordered conformation, while the other Q−1 identical values describing the disordered. In brief, *Q* has the meaning of the phase space volume of the problem. The energy of interaction is assumed to be different from zero when both $\gamma_{i}$ and $\gamma_{i+1}$ are equal to 1. The corresponding spin Hamiltonian is
(1)$$-\beta H_{Zimm–Bragg}\left(\left\{\gamma_{i}\right\}\right)=J\sum_{i=1}^{N}\delta \left(\gamma_{i},1\right)\delta \left(\gamma_{i+1},1\right).$$

Here J=βU, U(>0) is the absolute value of hydrogen bonding energy between the neighboring repeat units, $\beta =1/\kappa_{B}T$ is the inverse thermal energy and δ’s stand for the Kronecker symbols. For convenience we set $\kappa_{B}=1$ throughout the paper. We address the interested reader to Ref. [13] for detailed comparison of the Potts-like model defined by Equation (Equation 1) with the classical 1D Potts model.

The standard transfer matrix technique [14,15] results in the partition function
(2)$$Z_{Zimm–Bragg}\left(W,Q\right)=\sum_{\left\{\gamma_{i}\right\}}\prod_{i=1}^{N}{\left(M\right)}_{\gamma_{i},\gamma_{i+1}},$$
where ${\left(M\right)}_{\gamma_{i},\gamma_{i+1}}$ are the elements of the Q×Q matrix
(3)$$M\left(Q\times Q\right)=\left(\begin{array}{cccc} e^{J} & 1 & \cdots & 1\\ 1 & 1 & \cdots & 1\\ \cdots & \cdots & \cdots & \cdots \\ 1 & 1 & \cdots & 1 \end{array}\right).$$

Since there are only two linearly independent rows or columns in M(Q×Q), the order of the non-trivial part of the characteristic equation is also equal to two. Indeed [9], the eigenvalue problem $|\hat{M}-\hat{I}\Lambda |=0$, with the help of elementary transformations (which do not alter the determinant) reduces to
(4)$$\Lambda^{Q-2}\times \det\left(\begin{array}{cc} e^{J}-1-\Lambda & e^{J}-1\\ 1 & Q-\Lambda \end{array}\right)=0.$$

Neglecting the Q−2 trivial eigenvalues leads to the characteristic equation
(5)$$\Lambda^{2}-\Lambda \left(W-1+Q\right)+\left(W-1\right)\left(Q-1\right)=0,$$
where $W=e^{J}$. Resulting two roots are:(6)$$\Lambda_{1,2}=\frac{1}{2}\left[W-1+Q\pm \sqrt{{\left(W-Q+1\right)}^{2}+4\left(Q-1\right)}\right].$$

The partition function in Equation (Equation 2) can be simplified to read:(7)$$\begin{array}{c} Z\left(W,Q\right)=C_{1}\Lambda_{1}^{N}+C_{2}\Lambda_{2}^{N}=\Lambda_{1}^{N}\left[C_{1}+C_{2}{\left(\frac{\Lambda_{2}}{\Lambda_{1}}\right)}^{N}\right]=\\ \Lambda_{1}^{N}\left[C_{1}+C_{2}e^{-N/\xi }\right], \end{array}$$
where $C_{1}=\frac{Q-\Lambda_{2}}{\Lambda_{1}-\Lambda_{2}}$, $C_{2}=\frac{\Lambda_{1}-Q}{\Lambda_{1}-\Lambda_{2}}$ and
(8)$$\xi \left(W,Q\right)=1/\ln\left(\Lambda_{1}/\Lambda_{2}\right)$$
is the spatial correlation length, describing the characteristic scale of correlations in the model. In the model defined by Equation (Equation 1), the H-bond formation between two nucleotides takes place, when they both are in the same ordered conformation (γ=1). The order parameter (helicity degree) can be therefore defined as the average relative number of H-bonds as
(9)$$\begin{array}{c} \theta \left(W,Q\right)=\frac{<\sum_{i=1}^{N}\delta \left(\gamma_{i},1\right)\delta \left(\gamma_{i+1},1\right)>}{N}=\frac{1}{N}\frac{\sum_{\left\{\gamma_{i}\right\}}e^{J\sum_{i=1}^{N}\delta_{i}^{\left(2\right)}}\sum_{i=1}^{N}\delta_{i}^{\left(2\right)}}{Z}=\frac{1}{N}\frac{\partial \ln Z}{\partial \ln W}. \end{array}$$

Presented above are major formulas resulting from the Potts-like Hamiltonian model [9], defined by Equation (Equation 1).

### 2.2. Mapping the Potts-Like Spin Language to Zimm–Bragg Model

Originally, the Zimm–Bragg model is formulated with the help of rules to assign statistical weights per repeat unit for helices, coils and helix initiation sites and starts from the characteristic equation. A simple change of variables
(10)λ→Λ/Q;s→(W−1)/Q;σ→1/Q
in Equation (Equation 4) converts the Potts-like spin language to Zimm–Bragg model by yielding the well-known characteristic equation
(11)$$\det\left(\begin{array}{cc} s-\lambda & s\\ \sigma & 1-\lambda \end{array}\right)=\lambda^{2}-\lambda \left(s+1\right)+s\left(1-\sigma \right)=0,$$
with roots
(12)$$\lambda_{1,2}\left(s,\sigma \right)=\frac{1}{2}\left[1+s\pm \sqrt{{\left(1-s\right)}^{2}+4\sigma s}\right].$$

Therefore, the Hamiltonian in Equation (Equation 1) provides exactly the same thermodynamics of the Zimm–Bragg model and, hence, can be considered equivalent to it. Interesting enough, Equation (Equation 10) indicates that the parameters *W* and *Q* are independent from each other, while $s=\frac{W-1}{Q}$ and $\sigma =\frac{1}{Q}$ are not. It is straightforward to see that $s=\frac{W-1}{Q}=\frac{W}{Q}-\frac{1}{Q}$ or $s+\sigma =\frac{W}{Q}$ and ln(s+σ)=lnW−lnQ=βU−lnQ=βΔG, where ΔG is the free-energy cost of helix formation in a single repeat unit [4,9]. In other words, the {W,Q} parametrization operates with the enthalpic and entropic contributions separately, while the {s,σ} is related to the free energy and the entropic cost, which are related. We will use both parametrisations, freely converting from one to another using Equation (Equation 10) upon necessity.

### 2.3. Eigenvalue Analysis of the Zimm–Bragg Model

Since the Thermodynamics of the problem results from the partition function, which is entirely defined by chain length *N* and the eigenvalues $\lambda_{1}$ and $\lambda_{2}$, let’s first study the behavior of eigenvalues on helix propagation parameter *s*.

From Equation (Equation 12) and Figure 1a, $\lambda \underset{s\to \infty }{\to }\left\{1,s\right\}$ (actually, same as in σ→0 limit). Equation (Equation 11) can be rewritten to the iterative form λ=f(λ,s,σ) in two different ways, resulting in two asymptots:(13)$$\begin{array}{c} \lambda =1-\frac{\sigma s}{s-\lambda }\underset{\lambda \to 1}{\to }1-\sigma \\ \lambda =s-\frac{\sigma s}{1-\lambda }\underset{\lambda \to s}{\to }s+\sigma . \end{array}$$

At $s^{*}=1-2\sigma$, where the two asymptots cross, also the distance between the two roots has a minimum and $\lambda_{1}-\lambda_{2}=\Delta \lambda \left(s^{*}\right)=2\sqrt{\sigma (1-\sigma )}\approx 2\sqrt{\sigma }$. The expression for the correlation length (Figure 1b)
(14)$$\xi \left(\sigma ,s\right)=\ln^{-1}\left(\frac{\lambda_{1}}{\lambda_{2}}\right)=\ln^{-1}\left(\frac{1+s+\sqrt{{\left(1-s\right)}^{2}+4\sigma s}}{1+s-\sqrt{{\left(1-s\right)}^{2}+4\sigma s}}\right)$$
at the same $s^{*}=1-2\sigma$ point has a maximum and is equal to
(15)$$\xi \left(\sigma ,s^{*}\right)=\xi_{max}=\ln^{-1}\left(1+2\sqrt{\sigma -\sigma^{2}}\right)\approx \frac{1}{2\sqrt{\sigma }},$$
where we have assumed σ≪1 and used the ln(1+x)≈x approximation. As shown in the inset to Figure 1b, the approximation Equation (Equation 15) works very well and is getting better for smaller σ.

## 3. Results

The application of the mapping from Equation (Equation 10) to the partition function in Potts-like formulation Equation (Equation 7) results in the most general expression for the partition function of the Zimm–Bragg model [2] in {s,σ} parametrisation:(16)$$\begin{array}{c} Z\left(s,\sigma \right)=c_{1}\lambda_{1}^{N}+c_{2}\lambda_{2}^{N}=\lambda_{1}^{N}\left[c_{1}+c_{2}{\left(\frac{\lambda_{2}}{\lambda_{1}}\right)}^{N}\right]=\lambda_{1}^{N}\left[c_{1}+c_{2}e^{-N/\xi }\right], \end{array}$$
with $c_{1}=\frac{1-\lambda_{2}}{\lambda_{1}-\lambda_{2}}$, $c_{2}=\frac{\lambda_{1}-1}{\lambda_{1}-\lambda_{2}}$. Here
(17)$$\xi \left(s,\sigma \right)=1/\ln\left(\lambda_{1}/\lambda_{2}\right)$$
is the spatial correlation length, that can be straightforwardly obtained from the characteristic equation, but is for unknown reasons very rarely considered within the Zimm–Bragg analysis. It plays the role of relevant spatial scale and is directly related to persistence length of a polymer chain. As shown in Figure 1, the correlation length is a curve with maximum, which is finite, since no phase transition in 1D models with short-range interactions can take place (Pierls-Landau Theorem [16]).

Equation (Equation 16) represents exact expression, to which no assumptions about chain length has been applied yet, and we will use it as the starting point for the derivation of different size limits of the Zimm–Bragg model.

### 3.1. Size Limits of Partition Function

While the eigenvalues and the correlation length do not depend on chain length *N*, the partition function in Equation (Equation 16) (and consequently, thermodynamic potentials and the order parameter) does. Equation (Equation 16) can have there three size-dependent limits: N→∞, N≫ξ and N<ξ. Let’s consider them in order. It makes sense to perform analysis at transition point (for infinite *N*) $s^{*}=1-2\sigma$, where the correlation length is maximal.

#### 3.1.1. Infinite Chain Limit

First and the most obvious limit
(18)$$Z≃\lambda_{1}^{N};\;N\to \infty$$
is valid for the infinitely long chain and practically starts to work when there is at least a polymer of thousand persistence lengths (see Figure 2). This is the basic limit, taken by most theories and leading to simplest possible expressions for the order parameter. In Figure 2 all the presented curves are considered in units of Equation (Equation 18), which can also be considered as an asymptot for both the exact and long chain length (N≫ξ) expressions.

#### 3.1.2. Long Chain Limit

When the chain contains many persistence lengths, the partition function can be approximated as
(19)$$Z≃c_{1}\lambda_{1}^{N}=\frac{1-\lambda_{2}}{\lambda_{1}-\lambda_{2}}\lambda_{1}^{N};\;N\gg \xi .$$

As it is shown in Figure 2, the limit starts to work when the chain contains at least ten persistence lengths and thus can describe polymers.

#### 3.1.3. Short Chain Limit or Single Sequence Approximation

By its definition, the N<ξ condition refers to an oligomer, rather than to a polymer. For historical reasons the applicable approximation is also known as the single-sequence approximation of the Zimm–Bragg model. At the heart of the single-sequence approximation is the impossibility of having more than one uninterrupted sequence of helical (ordered) repeat units due to small system sizes (N<ξ). In the original formulation of the Zimm and Bragg [1] the probability of having disordered, coil conformation at a position in the middle of the chain has been considered negligible, i.e.,
(20)$$\frac{\sigma s}{{\left(1-s\right)}^{2}}\ll 1.$$

Naturally, Equation (Equation 20) plays the role of the small parameter, so that Equation (Equation 12) can be resolved into the Taylor series. If only the first term is kept, we obtain the eigenvalues
(21)$$\lambda_{1}\left(\sigma ,s\right)=1+\frac{\sigma s}{1-s};\;\lambda_{2}\left(\sigma ,s\right)=s-\frac{\sigma s}{1-s}.$$

After inserting the last expression into the Equation (Equation 16) we obtain:(22)$$Z\left(\sigma ,s\right)=\frac{\left(1-s+\frac{\sigma s}{1-s}\right){\left(1+\frac{\sigma s}{1-s}\right)}^{N}+\frac{\sigma s}{1-s}s^{N}{\left(1-\frac{\sigma }{1-s}\right)}^{N}}{1-s+\frac{2\sigma s}{1-s}}.$$

Resolving the powers into series, rearranging the results and keeping only terms linear in σ, we arrive at
(23)$$Z\left(\sigma ,s\right)=1+\frac{\sigma s}{{\left(1-s\right)}^{2}}\left(N-1-Ns+s^{N}\right)+O\left(\sigma^{2}\right).$$

The last is the famous single-sequence approximation formula [1,8,17,18], obtained from the partition function of the Zimm–Bragg model by taking the limits of the eigenvalues for the first time. As one can see in Figure 2, the approximation holds true up to chain length N≈2ξ, which fact was used in the recent application of zipper model [4]. Equation (Equation 23) is also known as the partition function of zipper model, as first shown by Schellman [17] in 1958, Gibbs and DiMarzio [19] in 1959 and later by Kittel [20] in 1969.

### 3.2. Exact Solution and Measurable Quantities

The order parameter (helicity degree) follows from the application of Equation (Equation 10) to Equation (Equation 9):(24)$$\begin{array}{c} \theta \left(s,\sigma \right)=\frac{1}{N}\frac{\partial \ln Z}{\partial \ln s}\frac{\partial \ln s}{\partial \ln W}=\frac{1}{N}\frac{\partial \ln Z}{\partial \ln s}\frac{s+\sigma }{s}=\theta_{Zimm–Bragg}\left(s,\sigma \right)\frac{s+\sigma }{s}, \end{array}$$
which obviously differs from the original definition
(25)$$\theta_{Zimm–Bragg}\left(s,\sigma \right)=\frac{1}{N}\frac{\partial \ln Z}{\partial \ln s}$$
by the $\frac{s+\sigma }{s}$ factor, which is very close to 1 for small σs.

This discrepancy arises due to the absence of Hamiltonian in the original formulation of Zimm–Bragg model. In brief, since each helical repeated unit contributes one *s* to the partition function, it was originally assumed that the average number of helical repeated units may be found from differentiating the partition function with respect to *s*. In other words, if $Z\approx \sum s^{N\theta }\sigma^{N\nu }$ (ν is the average number of uninterrupted helical segments), then θ=(1/N)dlnZ/dlns. Such approach cannot be true in general, since also the prefactors in the partition function(the number of ways to distribute the given number of helical units) depend on *s* (see [2,21] for details). Because of the better reasoning behind, we will stick to Equation (Equation 24) throughout the paper as a default definition of order parameter. Same result follows from Equation (Equation 24) with the exact partition function Equation (Equation 16), valid for any *N*. In Figure 3 the helicity degree from Equation (Equation 24) is plotted. We see that at *N* comparable to the correlation (or persistence) length, helicity degree doesn’t reach its limiting value at unity at any reasonable value of *s*, thus making it impossible to use the θ=0.5 condition to determine the transition point. An operational definition of transition point $s^{*}$ was used in [8], as the point where the behavior of θ(s) changes from the concave to convex, i.e., we search for an inflection point:(26)$$\theta^{\prime \prime }\left(s^{*}\right)=0.$$

Such definition of $s^{*}$ makes sense, since at infinite *N* it corresponds to the θ=0.5 condition. Transition point describes the stability of the system, and the transition interval
(27)$$\Delta s=\theta^{\prime }{\left(s\right)|}_{s^{*}}^{-1}$$
is related to the cooperativity. The conversion from *s* to temperatures is again made with the help of mapping in Equation (Equation 10) and the fact that W=exp(U/T):(28)$$\frac{T_{m}}{U}=\frac{1}{\ln\left(1+s^{*}/\sigma \right)};\;\frac{\Delta T_{m}}{U}=\frac{\Delta s{\left(T_{m}/U\right)}^{2}}{s^{*}+\sigma }.$$

The behavior of transition temperature and interval from Equation (Equation 28) is shown for different chain lengths in Figure 4.

## 4. Discussion

We compare all the limits of partition function to its exact expression Equation (Equation 16). Figure 2 reveals that the short chain or single-sequence approximation of partition function Equation (Equation 23) beyond the chain lengths of about $N\approx 2\xi_{max}$, looses its meaning and starts behaving in a way, very different from the exact partition function. From the same Figure 2 we see, that the range of validity of long chain limit of partition function Equation (Equation 19) does not overlap with that of the short chain limit. There is a gap in exactly the experimentally relevant range of two to five persistence lengths, where only the exact expression Equation (Equation 16) works. Above ten persistence lengths the long chain limit works very well, as expected. As to the chain limit ($Z=\lambda_{1}^{N}$) on the same Figure 2, it is used as a reducing factor, and is plotted as a constant line at 1. Interesting is the fact that the infinite chain limit works after several hundreds of persistence lengths.

The helicity degree in Figure 3 drawn according to Equations (Equation 24) and (Equation 16), illustrates that for short chains it never reaches 100% of helicity for any reasonable *s*. It has a simple logical explanation: for $N<\xi_{max}$ the final size effects heavily distort the end monomers. Helicity degree is less symmetrical and the inflection point deviates from θ=0.5 point.

Figure 4a elucidates that the transition (temperature) interval reaches its limiting value at about 500 persistence lengths and the curves for different σs overlap. As to the transition temperature, it reaches saturation at the scale of 10 persistence lengths. The larger a σ is, the sooner the saturation takes place.

The Hamiltonian formulation of the Zimm–Bragg model allowed to reconsider and compare the different limits of model partition function using the same language. As a result, an important conclusion has been drawn about the application of short chain approximation which must be heavily controlled by comparing chain length with the second size scale, the correlation or persistence length. The gap in approximations in the experimentally relevant range between 2 to 5 persistence length has been observed, where only the exact expression holds true. Besides, the definition of helicity degree has been corrected. In addition, it is shown, that the transition temperature is less sensitive to chain lengths, as opposed to the transition interval, which requires hundreds of persistence lengths to reach its asymptotic value. Since the Zimm–Bragg model finds its application in numerous experimental situations, related to the conformational transitions in Biopolymers, the summarized results are of general nature and are relevant for any such studies. 

## Figures and Tables

**Figure 1 polymers-13-01985-f001:**
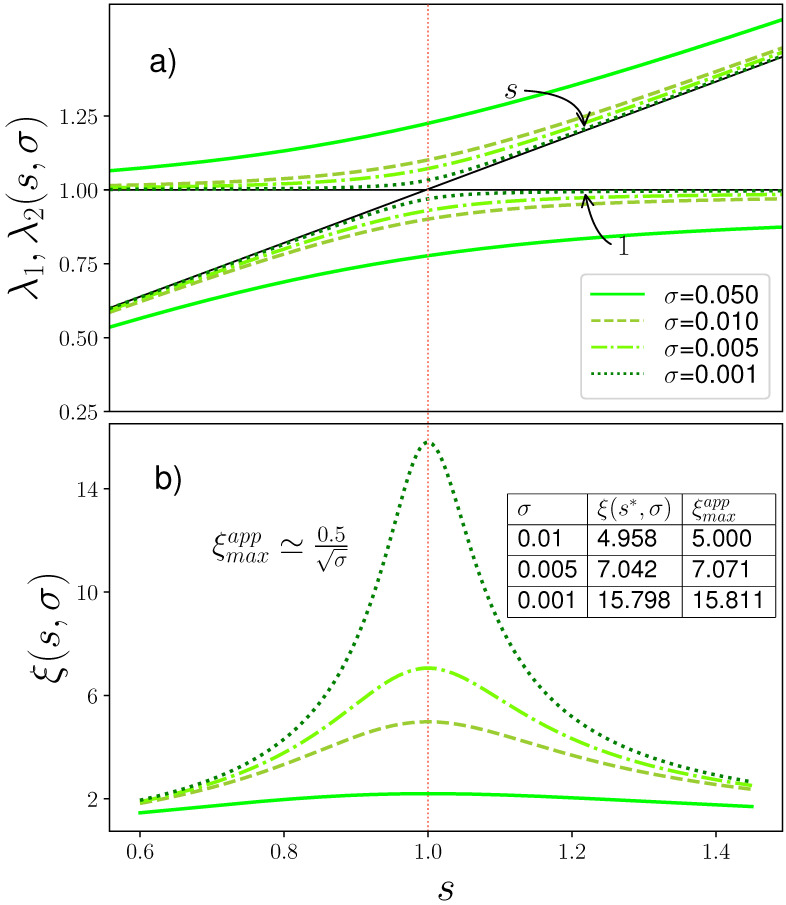
**Eigenvalue analysis for Zimm–Bragg model** over a range of stability parameter *s* at different values of cooperativity parameter σ. Transition point can be approximated as $s^{*}=1$. (**a**) Eigenvalues (defined by Equation (Equation 12)) have asymptots s+σ and 1−σ (not shown), the black lines shown instead visualize limiting (at σ→0) asymptots *s* and 1. σ values and the color codes shown in the inset are valid throughout the picture. (**b**) Spatial correlation length ξ (defined by Equation (Equation 17)) is a curve with maximum, that scales as $\xi_{max}^{app}≃\frac{0.5}{\sqrt{\sigma }}$; inset shows the comparison between exact and approximate results.

**Figure 2 polymers-13-01985-f002:**
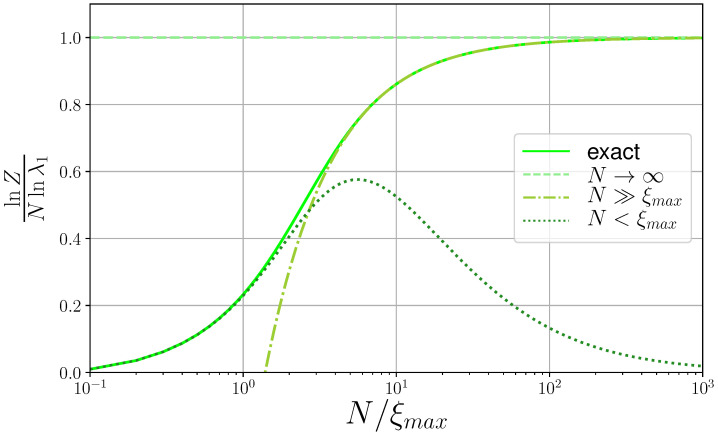
**Limits of partition function of the Zimm–Bragg model** and their size-scaling behavior. Partition function is in logarithmic units and is reduces to the infinite chain approximation; chain length is reduced to the correlation length and is plotted in logarithmic scale. All plotted formulas are at transition point $s^{*}=1-2\sigma$; σ=0.001, resulting in $\xi_{max}=15.8$. Legend is shown as the inset.

**Figure 3 polymers-13-01985-f003:**
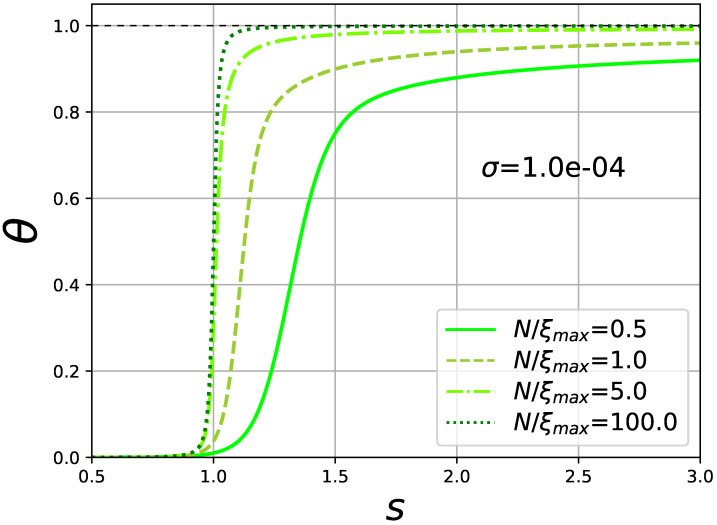
**Helicity degree θ** over a range of stability parameter *s* at different values of reduced chain lengths $N/\xi_{max}$; cooperativity parameter σ=0.0001 is same for all the curves.

**Figure 4 polymers-13-01985-f004:**
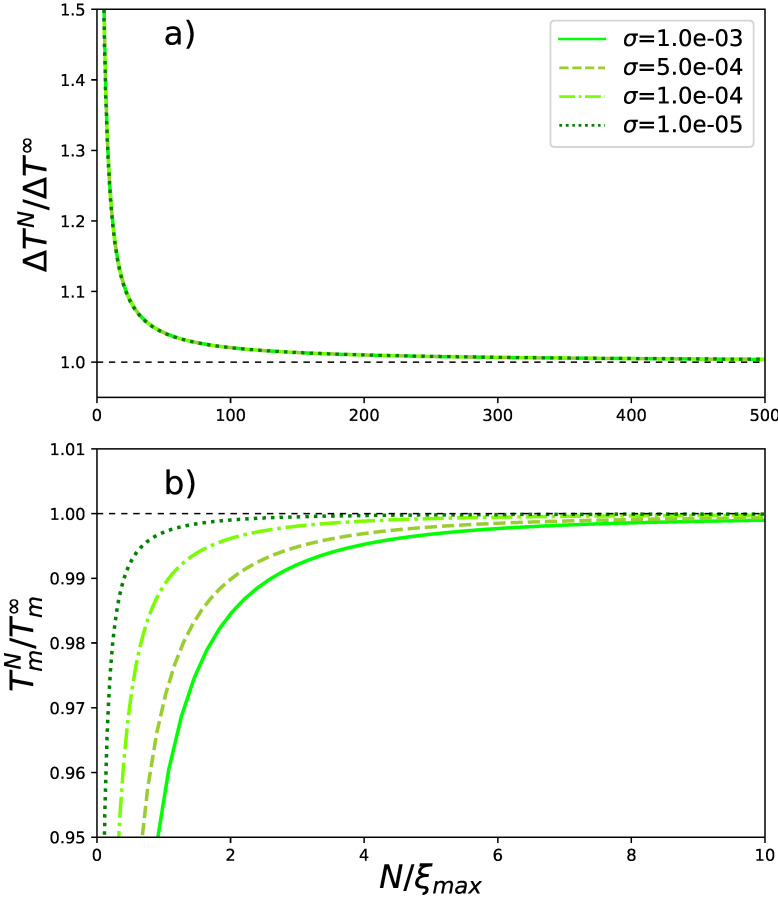
**Transition interval and temperature** in relative units: (**a**) $\Delta T^{N}/\Delta T^{\infty }$ and (**b**) $T_{m}^{N}/T_{m}^{\infty }$ over a range of reduced chain lengths $N/\xi_{max}$ at different values of cooperativity parameter σ (values shown in the inset).

## Data Availability

Not applicable.

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
