# Peer review of "System Size Dependence in the Zimm–Bragg Model: Partition Function Limits, Transition Temperature and Interval"

_polymers, 2021, doi:10.3390/polym13121985_

Round 1

Reviewer 1 Report

Review on: System size dependence in Zimm-Bragg model: Partition function limits, transition temperature and interval by Artem Badasyan submitted to Polymers.

In this article, the author employs the recently developed Hamiltonian formulation of Zimm-Bragg model to investigate size-dependent approximations of the model partition function. The author compares the chain length N with the maximum persistence length ξ of helical formation. Furthermore, the author claims to have derived the zipper model's partition function as limits of Zimm-Bragg model. Moreover, the author concludes that for a single chain and the long-chain approximations, critical considerations have shown a gap in the description for the range of experimentally relevant chain lengths of 5-10 persistence lengths.

In addition, the author has reported a correction to the helicity degree expression. Also, the author reports that at N/ξ=10, the transition temperature reaches its asymptotic behavior of infinite N.

These results suggest that these results contribute to the development of the Zimm-Bragg model, and they can be relevant to a wide range of applications, such as Biosensing and Biotechnologies. Therefore, I would recommend this article for publication in Polymers after the author considers the comments and suggestions submitted in a separate file.

Author Response

Reply to Reviewer 1

First, let me express my gratitude to the Reviewer for careful reading of my manuscript and the interesting remarks made.

As a general trend, I have noticed that all 4 Comments and Suggestions relate to the discussion of importance of different factors/interaction in DNA, while the goal of my paper is a general, abstract consideration of Zimm-Bragg model. In 2020 together with my colleagues we have shown, that the zipper model (aka single-sequence approximation or oligomer limit) can be derived as a limit from the Zimm-Bragg model, and have demonstrated its application to describe the oligomer DNA adsorption on carbon nanotubes (CNT). 

To make it clear, we didn't make any statement that Zimm-Bragg can be applied to the description of long DNA adsorption on CNT, since we are aware of several problems (including those 4 kindly mentioned by the Reviewer) that need to be accounted for before such application. Since most of papers report the application of short (actually, oligomer) DNAs for applications, we have limited ourself to this limit too.

I would say, it would be interesting to discuss these comments in a form of a Comment to the PRE 2020 paper (Tonoyan, Sh., Khechoyan, D., Mamasakhlisov, Ye., Badasyan, A. Statistical mechanics of DNA-nanotube adsorption. Phys. Rev. E. 101, 062422:1–5 (2020)). I encourage the Reviewer to write such a Comment, to which we will be glad to answer in details.

Although interesting, I do not see the abovementioned comments being directly related to the current manuscript, which is devoted to size-dependent analysis of Zimm-Bragg model in general. There are several applications of Zimm-Bragg model mentioned in manuscript, to underline, stress the importance of the model as a whole and nothing more.

To make my intention clear, I have added a sentence in the Introduction, clearly stating, that (lines 40-43):

"While in the current study there is no intention to apply the Zimm-Bragg model to describe the 
adsorption of long ssDNA to CNT, better understanding of different 
limits of the model is a first step towards possible future 
developments in this direction."

Having said that, I will try to answer the questions raised, as much as I see them related to the current study.

Reply to Comments and Suggestions

Reviewer said: 
" 1) A successful model for studying the flexibility of DNA is the worm-like chain model that describes
the inherent flexibility of polymer in terms of the persistence length for long scale polymer chains.
Therefore, the author should compare the two models in the scale of long-chain polymers or
comment on that in the Introduction section. 
"

Author replies:
I accept the comment in the sense, that some text has to be ammended to better explain the approach. 

At the same time I have to say it explicitly, neither the flexibility of DNA is considered in any part of the paper, nor the worm-like chain model. In fact, there is no cartesian size measure in the Zimm-Bragg model, everything is in terms of the number of repeat units. The two models are incomparable, since they describe different features of polymers. To avoid the possible misunderstanding that may arise with the readers, I have added an explanation, that size is measured in repeat units. Please see the second paragraph of Introduction (lines 22-26):

"Speaking about size in terms of Zimm-Bragg model, it is 
relevant to mention that the distance is measured in numbers of repeating units and the conformation is described with spins. Therefore, the questions about polymer coil size cannot be answered within the Zimm-Bragg model. In this respect it is similar to the Ising and Potts spin models \cite{baxter}, which also contain no measure of distance and do not describe the extension of a crystal."

Reviewer said:
" 2) The literature has argued that two main forces lead to the observed stiffness of the polymer chains
(such as DNA), namely the electrostatic repulsion between the phosphate groups on the backbone
and the π-stacking interactions between adjacent nucleotides. The author should comment on how
Zimm-Bragg model could be extended to count for these effects."

Author replies:
While the comment is correct, it does not relate to the goal of the paper, so I cannot implement it. Let me explain.
In this particular paper I do not discuss the application of Zimm-Bragg model to describe DNA conformations. The discussion regarding the origins of DNA stiffness, as suggested by the Reviewer, will drive the focus of the reader out of the goal of the current paper: the formal consideration of Zimm-Bragg model properties. 
I have considered the problem of presence of two interactions with different interaction scales in view of DNA melting (https://doi.org/10.1002/bip.20143), but the topic has nothing to do with the current paper.

Reviewer said:

"3) Furthermore, solvent effects, such as counter ions around the DNA, can play an essential role in
screening the electrostatic interactions and so in the flexibility of the polymer chain. Finally, the
author should comment on how this neutralization could increase flexibility."

Author replies:
While the comment is correct, it does not relate to the goal of the paper, so I cannot implement it. Let me explain.
I do not see how the screening affecting DNA flexibility is related to the current paper, which is devoted to the formal consideration of Zimm-Bragg model, on how its properties depend on the degree of polymerization. 
If, in the future, I decide to adapt the Zimm-Bragg model to describe DNA conformations, all 4 factors mentioned above should undoubtedly be addressed. In fact, there is also at least one more relevant factor, that needs to be considered for DNA: the loop entropy. But all these are out of the focus of the current paper.

Reviewer said:

"4. Moreover, the persistence length of the polymer chain, such as DNA, could be sequence-dependent.
The author should comment on how this could be included in the presented model."

Author replies:
Comment is accepted and implemented.

The account of sequence-dependence was considered in (Badasyan, A. V., Grygoryan, A. V., Mamasakhlisov, E. Sh., Benight, A. S. & Morozov, V. F. The helix-coil transition in heterogeneous
double stranded DNA: Microcanonical method. J. Chem. Phys. 123, 194701:1–6 (2005).) 
Following your advice, I have added some text on lines 73-76: 
"... and for the account of 
structural heterogeneity (heteropolymer) \cite{badasyan2005}. Since the
most of applications of Zimm-Bragg model are devoted to the polypeptide 
conformations and make use of a homopolymeric version of the model, I 
will undertake the same strategy."

Reply to Some minor questions/comments

Reviewer said:
"1) ... , but the sign does not match on both sides. Can the author comment
on that?"

Author replies:
Comment is accepted and implemented.

The minus on the left is from the Boltzmann factor, and to the right I have to write the energy of hydrogen bonding times number of pairs (the Hamiltonian) times (-\beta). The energy is negative, since hydrogen bonding is attaractive. It is easier to write the energy as (-U), where U>0, then the two minuses will eliminate each other and we are left with a positive parameter U. I have re-written the explanatory part to:
"... is the absolute value of hydrogen bonding energy between the neighboring repeat units." see lines 79-81.

Reviewer said:
"2) In Eq. (17), there should be ``ln" instead of ``log" to be consistent with other formulas."

Author replies:
Accepted, change implemented in Eq.(8) and Eq.(17).

Reviewer said:
"The author might also have to explain the thermodynamic relationship between the variables s and
σ..."

Author replies:
Comment accepted and implemented.

We have several times discussed this issue in our previous publications (see, e.g. Ref[4,9]). Since it is found relevant by the Reviewer, we are glad to mention it once more in the current paper. Before the section 2.3 (lines 91-95) I have added some text:
"
It is straightforward to see that 
$s=\frac{W-1}{Q}=\frac{W}{Q}-\frac{1}{Q}$ or $s+\sigma=\frac{W}{Q}$ and 
$\ln(s+\sigma)=\ln W-\ln Q=\beta U-\ln Q=\beta \Delta G$, where $\Delta 
G$ is the free-energy cost of helix formation in a single repeat unit \cite{shush2020,pre10}. In other 
words, the $\{W,Q\}$ parametrization operates with the enthalpic and 
entropic contributions separately, while the $\{s,\sigma\}$ is related 
to the free energy and the entropic cost, which are related.
"

Reviewer said:
"4) Just before the Eq. (28), in the expression
, the author may miss the Boltzmann's constant...
."

Author reply:
Comment accepted and implemented. 

Indeed it introduces some inconsistency. Since no direct comparison is made with the real data, during the calculations we set the Boltzmann constant equal to unity. We have added the following sentence on the line 82:
"For convenience we set $\kappa_B=1$ throughout the paper."

Thank you for your detailed consideration and useful comments. I have addressed all of your comments and have implemented most of them, providing the explanation on those points, which I found out of the focus of the paper. Hope in its current form you will find my paper publishable.

Reviewer 2 Report

The proposal of a statistical mechanics model for partition function limits, transition temperature and interval. It is a good job where the different variables were analyzed. It is a well posed article, with interesting results. I suggest be accepted.

Author Response

Reply to Reviewer 2

Thank you for understanding the goals of the paper and for your positive opinion.